# Storage Efficient and Dynamic Flexible Runtime Channel Pruning via Deep Reinforcement Learning

## Abstract

In this paper, we propose a deep reinforcement learning (DRL) based framework to efficiently perform runtime channel pruning on convolutional neural networks (CNNs). Our DRL-based framework aims to learn a pruning strategy to determine how many and which channels to be pruned in each convolutional layer, depending on each specific input instance at runtime. The learned policy optimizes the performance of the network by restricting the computational resource on layers under an overall computation budget. Furthermore, unlike other runtime pruning methods which require to store all channels parameters for inference, our framework can reduce parameters storage consumption for deployment by introducing a static pruning component. Comparison experimental results with existing runtime and static pruning methods on state-of-the-art CNNs demonstrate that our proposed framework is able to provide a tradeoff between dynamic flexibility and storage efficiency in runtime channel pruning.

## 1 Introduction

In recent years, convolutional neural networks (CNNs) have been proven to be effective in a wide range of computer vision tasks, such as image classification (Krizhevsky et al., 2012; Simonyan & Zisserman, 2015; He et al., 2016), objection detection (He et al., 2017; Zhou et al., 2019; Law & Deng, 2018), segmentation (He et al., 2017; Zhu et al., 2019). Therefore, nowadays, many computer-vision-based systems, such as automatic-driving cars, security surveillance cameras, and robotics, are built on the power of CNNs. However, since most state-of-the-art CNNs require expensive computation power for inference and huge storage space to store large amount of parameters, the limitation of energy, computation and storage on mobile or edge devices has become the major bottleneck on real-world deployments of CNNs. Existing studies have been focused on speeding up the execution of CNNs for inference on edge devices by model compression using matrix decomposition (Denil et al., 2013; Masana et al., 2017), network quantization (Courbariaux et al., 2016), network pruning (Dong et al., 2017), etc. Among these approaches, channel pruning has shown promising performance (He et al., 2017; Luo et al., 2017; Zhuang et al., 2018; Peng et al., 2019). Specifically, channel pruning discards an entire input or output channel and keep the rest of the model with structures.

Most channel pruning approaches can be categorized into two types: runtime approaches and static approaches. Static channel pruning approaches aim to design a measurement to evaluate the importance of each channel over the whole training dataset and remove the least important channels to minimize the loss of performance after pruning. By permanently pruning a number of channels, the computation and storage cost of CNNs can be dramatically reduced when being deployed, and the inference execution can be accelerated consequently. Runtime channel pruning approaches have been recently proposed to achieve dynamic channel pruning on each specific instance (Gao et al., 2019; Luo & Wu, 2018). To be specific, the goal of runtime approaches aims to evaluate the channel importance at runtime, which is assumed to be different on different input instances. By pruning channels dynamically, different pruned structures can be considered as different routing of data stream inside CNNs. This kind of approaches is able to significantly improve the representation capability of CNNs, and thus achieve better performance in terms of prediction accuracy compared with static approaches. However, previous runtime approaches trade storage cost off dynamic flex-

ibility of pruning. To achieve dynamic pruning on different specific instances, all parameters of kernels are required to be stored (or even more parameters are introduced). This makes runtime approaches not applicable on resource-limited edge devices. Moreover, most of previous runtime approaches only evaluate the importance among channels in each single layer independently, without considering the difference in efficiency among layers.

In this paper, to address the aforementioned issues of runtime channel pruning approaches, we propose a deep reinforcement learning (DRL) based pruning framework. Basically, we aim to apply DRL to prune CNNs by maximizing received rewards, which are designed to satisfy the overall budget constraints along side with network's training accuracy. Note that automatic channel pruning by DRL is a difficult task because the action space is usually very huge. Specifically, the discrete action space for the DRL agent is as large as the number of channels at each layer, and the action spaces may vary among layers since there are different numbers of channels in different layers. To facilitate pruning CNNs by DRL, for each layer, we first design a novel prediction component to estimate the importance of channels, and then develop a DRL-based component to learn the sparsity ratio of the layer, i.e., how many channels should be pruned.

More specifically, different from previous runtime channel pruning approaches, which only learn runtime importance of each channel, we propose to learn both runtime importance and additionally static importance for each channel. While runtime importance maintains the saliency of specific channels for each given specific input, the static importance captures the overall saliency of the corresponding channel among the whole dataset. According to each type of the channel importance, we further design different DRL agents (i.e., a runtime agent and a static agent) to learn a sparsity ratio in a layer-wise manner. The sparsity ratio learned by the runtime agent together with the estimated runtime importance of channels are used to generate runtime pruning structures, while the sparsity ratio learned by the static agent together with the estimated static importance of channels are used to generate static (permanent) pruning structures. By considering both the pruning structures, our framework is able to provide a trade-off between storage efficiency and dynamic flexibility for runtime channel pruning.

In summary, our contributions are 2-fold. First, we propose to prune channels by taking both runtime and static information of the environment into consideration. Runtime information endows pruning with flexibility based on different input instances while static information reduces the number of parameters in deployment, leading to storage reduction, which cannot be achieved by conventional runtime pruning approaches. Second, we propose to use DRL to determine sparsity ratios, which is different from the previous pruning approaches which manually set sparsity ratios. Extensive experiments demonstrate the effectiveness of our method.

## 2 RELATED WORK AND PRELIMINARY

### 2.1 STRUCTURE PRUNING

Wen et al. (2016) pioneered structure pruning in deep neural network by imposing the $L_{2,1}$ norm in training. Under the same framework, Liu et al. (2017) regarded parameters in batch normalization as channel selection signal, which is minimized to achieve pruning during training. He et al. (2017) formulated channel pruning into a two-step iterative process including LASSO regression based channel selection and least square reconstruction. Luo et al. (2017) formulated channel pruning as minimization of difference of output features, which is solved by greedy selection. Zhuang et al. (2018) further considered early prediction, reconstruction loss and final loss to select importance channels. Overall, structure pruning methods accelerate inference by producing regular and compact model. However, this brought regularness requires preserving more parameters to ensure performance.

### 2.2 DYNAMIC PRUNING

Dynamic pruning provides different pruning strategies according to input data. Wang et al. (2018) proposed to reduce computation by skipping layers or channels based on the analysis of input features. Luo & Wu (2018) proposed to use layer input to learn channel importance, which is then binarized for pruning. Gao et al. (2019) applied the same framework while extended features selection in both input and output features. Similarly, Liu & Deng (2018) introduced multiple branches

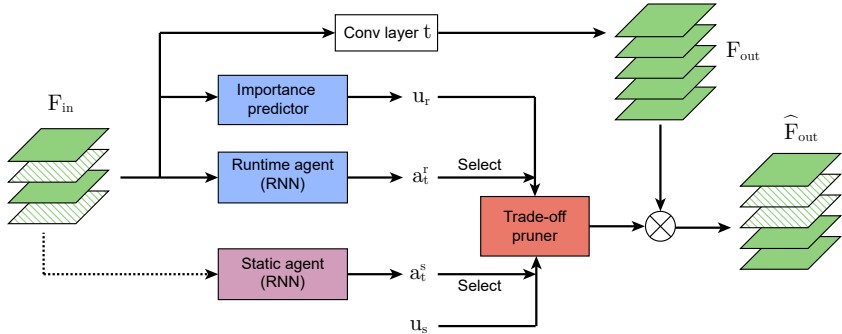

Figure 1: Illustration of our proposed DRL-based runtime pruning framework.

for runtime inference according to inputs. A gating module is learnt to guide the flow of feature maps. Bolukbasi et al. (2017) learned to choose the components of a deep network to be evaluated for each input adaptively. Early exit is introduced to accelerate computation. Dynamic pruning adaptively takes different actions for different inputs, which is able to accelerate the overall inference time. However, the original high-precision model needs to be stored, together with extra parameters for making specified pruning actions. Rosenbaum et al. (2018) proposed to learn routers to route layers output to different next layers, in order to adjust a network to multi-task learning.

### 2.3 DEEP REINFORCEMENT LEARNING IN PRUNING

Channel selection is on trial using deep reinforcement learning. Lin et al. (2017) trained a LSTM model to remember and provide channel pruning strategy for backbone CNN model, which is conducted using reinforcement learning techniques. He et al. (2018) proposed to determine the compression ratio in each layer by training an agent regarding the pruning-retraining process as an environment.

### 2.4 PRELIMINARY

**Reinforcement Learning** We consider a standard setup of reinforcement learning: an agent sequentially takes actions over a sequence of time steps in an environment, in order to maximize the cumulative reward (Sutton & Barto, 1998). This problem can be formulated as a Markov Decision Process (MDP) of a tuple $(\mathcal{S}, \mathcal{A}, \mathcal{P}, R, \gamma)$, where $\mathcal{S}$ is the state space, $\mathcal{A}$ is the action space, $\mathcal{P} : \mathcal{S} \times \mathcal{A} \times \mathcal{S} \rightarrow [0, 1]$ is transition probabilities, $R : \mathcal{S} \times \mathcal{A} \rightarrow \mathbb{R}$ is the reward function, and $\gamma \in [0, 1)$ is the discount factor. The goal of reinforcement learning is to learn a policy $\pi(a|s)$ that maximizes the objective of cumulative rewards over finite time steps,

$$\max_{\pi} \sum_{t=0}^{T} R(s_t, a_t),$$

where $s_t \in \mathcal{S}$ and $a_t \in \mathcal{A}$ are state and taken action at time step $t$ respectively.

## 3 DRL-BASED RUNTIME PRUNING FRAMEWORK

The overview of our proposed framework is presented in Fig. 1. To prune convolutional layer $t$, we learn two types of learnable *channel importance*: *runtime* channel importance $\mathbf{u}_r \in \mathbb{R}^{C \times 1}$ and *static* channel importance $\mathbf{u}_s \in \mathbb{R}^{C \times 1}$, where $C$ is the number of channels in layer $t$. The runtime channel importance $\mathbf{u}_r$ is generated by a subnetwork *importance predictor* $f(\cdot)$, which takes the input feature map $\mathbf{F}_{in}$ as input, while the static channel importance $\mathbf{u}_s$ is randomly initialized and updated during training. Both $\mathbf{u}_r$ and $\mathbf{u}_s$ indicate the channel importance of the full precision output feature map $\mathbf{F}_{out}$ through a convolution layer. Channels are selected to be pruned according to the values of each element in $\mathbf{u}_r$ and $\mathbf{u}_s$, and how many channels to be selected is decided by the sparsity ratios $d_r$ and $d_s$, respectively. To learn the sparsity ratios $d_r$ and $d_s$, two DRL agents, the *runtime agent* and the *static agent*, are introduced, where actions $a_t^r$ and $a_t^s$ are defined to set values of $d_r$ and $d_s$, respectively. The detail of the two DRL agents are described in Sec. 3.3. Consequently,

a *trade-off pruner* $g(\cdot)$ is performed to balance the runtime and static pruning results, and output a decision mask $\mathbf{M}$ of binary values (1/0) to indicate which channels to be pruned (1: pruned, 0: preserved), as well as a unified channel importance vector $\mathbf{u} \in \mathbb{R}^{C \times 1}$ as follows,

$$[\mathbf{M}, \mathbf{u}] = g(\mathbf{u}_r, \mathbf{u}_s, d_r, d_s). \tag{1}$$

The final output after pruning is constructed by multiplying the full precision output feature map $\mathbf{F}_{out}$, by $1 - \mathbf{M}$ and $\mathbf{u}$ as,

$$\hat{\mathbf{F}}_{out} = \mathbf{F}_{out} \otimes (\mathbf{1} - \mathbf{M}) \otimes \mathbf{u}, \tag{2}$$

where $\otimes$ is the broadcast element-wise multiplier, and $\mathbf{1}$ is the matrix of the same size as $\mathbf{M}$ with all the elements being 1. In the following, we introduce how to learn the runtime channel importance vector $\mathbf{u}_r$ and the static channel importance vector $\mathbf{u}_s$ in Sec. 3.1, how to construct the trade-off pruner $g(\cdot)$ in Sec. 3.2, and how to design the two DRL agents in Sec. 3.3.

### 3.1 LEARNABLE CHANNEL IMPORTANCE

We consider that a convolutional layer takes input of feature map $\mathbf{F}_{in} \in \mathbb{R}^{C_{in} \times H_{in} \times W_{in}}$ and generates an output feature map $\mathbf{F}_{out} \in \mathbb{R}^{C_{out} \times H_{out} \times W_{out}}$, where $C_*$, $H_*$ and $W_*$ are the number of channels, width and height of the feature map $\mathbf{F}_*$, respectively. Each element of the channel importance vectors $\mathbf{u}_r \in \mathbb{R}^{C_{out}}$ and $\mathbf{u}_s \in \mathbb{R}^{C_{out}}$ represents the importance value of the corresponding channel, respectively. In the following, we drop the subscript $_{out}$ for simplicity in presentation.

#### 3.1.1 RUNTIME CHANNEL IMPORTANCE

As mentioned above, the runtime channel importance $\mathbf{u}_r$ of output feature $\mathbf{F}_{out}$ is predicted by a *importance predictor* $f(\cdot)$, which takes $\mathbf{F}_{in}$ as input. Therefore, $\mathbf{u}_r$ can be considered as a function of $\mathbf{F}_{in}$, whose values vary over different input instances. In this paper, we design a subnetwork to approximate $f(\cdot)$, which is expected to be of a small size and computationally efficient. Similar to many existing dynamic network pruning methods (Gao et al., 2019; Hu et al., 2018; Luo & Wu, 2018), we use global pooling layer as the first layer in $f(\cdot)$, because global pooling is computationally efficient and it can reduce the dimension of $\mathbf{F}_{in}$ dramatically. We then feed the output of global pooling into a fully-connected layer without any activation function. The output of fully-connected layer is the runtime channel importance vector $\mathbf{u}_r$.

Which channels to be preserved / pruned at runtime are determined according to the values of $\mathbf{u}_r$. We denote by $\mathbf{M}_r \in \{0, 1\}^C$ a mask for pruning, where if the value is 0, then the corresponding channel is preserved, otherwise pruned. For now, suppose a sparsity ratio $d_r$ for runtime pruning has already been generated via the dynamic DRL agent, which will be introduced in Sec. 3.3. We then prune $(C - \lceil d_r C \rceil)$ channels with the smallest importance values in $\mathbf{u}_r$. Accordingly, the value of an element in $\mathbf{M}_r$ is set to be 1 if the corresponding channel is pruned, otherwise 0.

#### 3.1.2 STATIC CHANNEL IMPORTANCE

The static channel importance vector $\mathbf{u}_s$ is to capture the global information for pruning, and thus is learned from the whole dataset. It is randomly generated and learned through backpropagation. Similar to runtime channel pruning, given a sparsity ratio $d_s$ learned by the static DRL agent, $(C - \lceil d_s C \rceil)$ channels with smallest importance values in $\mathbf{u}_s$ are pruned, and a mask $\mathbf{M}_s \in \{0, 1\}^C$ is generated to indicate the static pruning results.

### 3.2 TRADE-OFF PRUNER

With the runtime and the static pruning decisions, $\mathbf{M}_r$ and $\mathbf{M}_s$, we now propose a *trade-off pruner* to generate a unified channel pruning decision. The main idea behind the trade-off pruner is to 1) prune those channels which are agreed to be pruned by both decisions, and 2) prune a portion of the rest channels by weighted votes from both decisions.

To be specific, we define the mask representing channels pruned by both decisions as

$$\mathbf{M}_o = \mathbf{M}_s \wedge \mathbf{M}_r, \tag{3}$$

where $\wedge$ is element-wise logical AND and 1/0 in mask represents logical *true* or *false*. The channels indicated to be pruned by $\mathbf{M}_o$ (i.e., the corresponding values are 1) are pruned in final. The channels

which are determined to be pruned by $\mathbf{M}_r$ but not by $\mathbf{M}_s$ can be represented by a new mask $\overline{\mathbf{M}}_r = \mathbf{M}_r - \mathbf{M}_o$. Similarly, the channels which are determined to be pruned by $\mathbf{M}_r$ but not by $\mathbf{M}_s$ can be represented by another new mask $\overline{\mathbf{M}}_s = \mathbf{M}_s - \mathbf{M}_o$.

To control the trade-off between $\mathbf{M}_r$ and $\mathbf{M}_s$, we define a rate $R_r$ denoting how much we trust the pruning decision made by $\mathbf{M}_r$, while $1 - R_r$ is for $\mathbf{M}_s$. That means the channels selected by $\overline{\mathbf{M}}_r$ will be finally pruned with the rate $R_r$. Specifically, the number of channels which are selected by $\overline{\mathbf{M}}_r$ and finally will be pruned is

$$C_r^{'} = \lfloor R_r(\mathbf{1}^\top \overline{\mathbf{M}}_r) \rfloor, \tag{4}$$

where $\mathbf{1}^\top \overline{\mathbf{M}}_r$ returns the number of channels selected by $\overline{\mathbf{M}}_r$. We then select the first $C_r^{'}$-smallest important channels which are recommended to be pruned by $\overline{\mathbf{M}}_r$ to form a mask $\widehat{\mathbf{M}}_r$. Similarly, for static pruning, we select the first $C_s^{'}$-smallest important channels which are recommended to be pruned by $\overline{\mathbf{M}}_s$ to form another mask $\widehat{\mathbf{M}}_s$, where $C_s^{'} = \lfloor (1 - R_r)(\mathbf{1}^\top \overline{\mathbf{M}}_s) \rfloor$.

The final trade-off pruning mask is defined as

$$\mathbf{M} = \mathbf{M}_o + \widehat{\mathbf{M}}_r + \widehat{\mathbf{M}}_s. \tag{5}$$

Moreover, in this work, the unified channel importance is simply defined as follows,

$$\mathbf{u} = \mathbf{u}_r \otimes \mathbf{u}_s. \tag{6}$$

With the trade-off pruning mask $\mathbf{M}$ and the unified channel importance $\mathbf{u}$, the pruned output feature $\hat{\mathbf{F}}_{out}$ can be generated by Eq. 2.

### 3.3 DRL BASED PRUNING

In this section, we present how to formulate the problems of learning the ratios $d_s$ and $d_t$ for static pruning and runtime pruning, as a MDP, and solve it via DRL, respectively.

#### 3.3.1 DRL FOR RUNTIME PRUNING

In the MDP for runtime pruning, we consider the $t$-th layer of the network as the $t$-th timestamp. The details of the MDP are listed as follows.

**State**  Given an input feature map $\mathbf{F}_{in}$ of layer $t$, we pass it to a global pooling layer to reduce its dimension to $\mathbb{R}^{C_{in}}$, where $C_{in}$ is the number of input channel of layer $t$. Since $C_{in}$ varies among layers, we feed the output of global pooling to a layer-dependent encoder to project it to a fix-length vector $s_t^r$, which is considered as as a *state* representation of DRL in the context of runtime pruning.

**Action**  The *action* $a_t^r$ is defined as the sparsity ratio at layer $t$, alternating $d_r$ in runtime pruning mentioned in Section 3.1.1. Existing DRL-base pruning method RNP (Lin et al., 2017) uses a unified discrete actions space with $k$ actions which are too coarse to achieve high accuracy. However, fine-grained discrete action space as large as number of channels suffers from exploration difficulty. Therefore, instead of using discrete action spaces, we propose a continuous action space with action $a_t^r \in (0, 1]$. To avoid over-pruning the filters and crashing in training, we set a minimum sparsity ratio $+\alpha$ such that $a_t^r \in (+\alpha, 1]$.

**Reward**  The reward function is proposed to consider both network accuracy and computation budget. We define the accuracy relative reward based on the loss of pruned backbone network,

$$R_{acc}^r = -\mathcal{L}_{CNN}, \tag{7}$$

where $\mathcal{L}_{CNN}$ is the loss in CNN, and it may vary in scale among different training stage, i.e. large at beginning of training and small near convergence. To avoid the instability brought by the reward scale, $R_{acc}^r$ is normalized by a moving average,

$$R_{acc}^{r'} = R_{acc}^r / \beta_b, \tag{8}$$

$$\beta_b = \lambda \beta_{b-1} + (1 - \lambda) R_{acc}^r, \tag{9}$$

where $\beta_b$ is the moving average at the $b$-th training batch and $\lambda$ is the moving weight.

To force computation of the pruned network under a given computation budget, we define a exponential reward function of budget regarding reward $R_{bud}^r$:

$$R_{bud}^r = \begin{cases} \exp(\alpha_1(B_{com} - \overline{B}_{com})) - 1, & B_{com} > \overline{B}_{com}, \\ 0, & \text{otherwise}, \end{cases} \tag{10}$$

where $B_{com}$ is the computation consumption, which is calculated based on the current of pruned strategy, and $\overline{B}_{com}$ is the given computation budget constraint. Finally we sum up the two rewards to form sparse rewards, with being non-zero at terminated step $T$ and zeros at other time step $t < T$,

$$R_t^r = \begin{cases} R_{acc}^{r'} + R_{bud}^r, & t = T, \\ 0, & t < T. \end{cases} \tag{11}$$

**Actor-Critic Agent**   To solve the continuous action space problem, we choose a commonly used actor-critic agent with a Gaussian policy. Actor-critic agent consists of two components: 1) actor outputs the mean and variance to form a Gaussian policy where the continuous action are sampled from; 2) critic outputs a scalar predicting the future discounted accumulated reward and assists the policy training. Actor network and Critic network share one-layer RNN which takes state $s_t^r$ as input. The output of RNN is fed into actor specific network constructed by two branches of fully-connected layers, leading to the mean and variance of the Gaussian policy. The action is sampled for the Gaussian distribution outputed by the actor:

$$a_t^r \sim \mathcal{N}(\mu(s_t^r; \theta^r), \sigma(s_t^r; \theta^r)), \tag{12}$$

where $\mu(s_t^r; \theta^r)$ and $\sigma(s_t^r; \theta^r)$ is the mean and variance outputed from actor network. The Critic specific network has one fully-connected layer after the shared RNN, and outputs the predictive value $V(s_t^r; \theta^r)$.

To optimize the actor-critic agent, Proximal Policy Optimization (PPO) (Schulman et al., 2017) is used. Note that we relax the action $a_t^r$ to $(-\infty, +\infty)$ in PPO, and use truncate function to clip $a_t^r$ in $(+\alpha, 1]$ when perform pruning.

Besides, an additional regularizer is introduced to restrict the relaxed $a_t^r$ staying in range $(+\alpha, 1]$,

$$\mathcal{L}_a = \frac{1}{2}||a_t^r - \max(\min(a_t^r, 1), +\alpha)||_2^2. \tag{13}$$

### 3.3.2   DRL FOR STATIC PRUNING

Similar to runtime pruning, the MDP in static pruning is also formulated layer-by-layer. The difference against runtime pruning is the definition of state and reward.

**State**   The state $s_t^s$ in static pruning is defined as the full shape of $\mathbf{F}_{out}$, and does not depend on $\mathbf{F}_{out}$ and the current input data.

**Action**   Action $a_t^s$ is sampled from actor's outputed Gaussian policy, and it is to alternate the sparsity $d_s$ in static pruning mentioned in Section 3.1.2.

**Reward**   The reward function takes both network accuracy and parameters budget into consideration. The accuracy relative is defined as the same as that in runtime pruning:

$$R_{acc}^s = R_{acc}^{r'}. \tag{14}$$

To reduce the number of parameters of network to satisfy the parameters storage budget, the parameters relative reward is defined in an exponential form as,

$$R_{param}^s = \begin{cases} \exp(\alpha_2(B_{param} - \overline{B}_{param})) - 1, & B_{param} > \overline{B}_{param}, \\ 0, & \text{otherwise}, \end{cases} \tag{15}$$

where $B_{param}$ is the number of preserved parameters after static pruning and $\overline{B}_{param}$ is the parameters storage budget.

| Method | Baseline acc. | Acc. | $\Delta acc.$ | Speed-up | #Params |
|---|---|---|---|---|---|
| FBS (Gao et al., 2019) | 91.37 | 89.88 | -1.49 | **3.93**$\times$ | 1.11$\times$ |
| RNP (Lin et al., 2017) | 92.07 | 84.93 | -7.14 | 3.56$\times$ | 1.00$\times$ |
| ours (runtime only $R_r = 1$) | 92.07 | 91.333 | **-0.737** | 3.92$\times$ | 1.31$\times$ |
| ours ($R_r = 0.5$) | 92.07 | 91.066 | **-1.004** | 3.92$\times$ | **0.78**$\times$ |

Table 1: Comparison to state-of-the-art runtime pruning methods on CIFAR-10 at sparsity 0.5. Speed-up is calculated on MACs.

| Method | Baseline acc. | Acc. | $\Delta acc.$ | Speed-up | #Params |
|---|---|---|---|---|---|
| FBS (Gao et al., 2019) | 91.37 | 91.23 | -0.14 | 2$\times$ | 1.11$\times$ |
| ours ($R_r = 1.0$) | 92.07 | 93.178 | **+1.108** | 1.99$\times$ | **1.31**$\times$ |
| ours ($R_r = 0.5$) | 92.07 | 92.502 | **+0.432** | 1.99$\times$ | **0.97**$\times$ |

Table 2: Comparison to state-of-the-art runtime pruning methods on CIFAR-10 at sparsity 0.7. Speed-up is calculated on MACs.

**Actor-Critic Agent**   This agent is similar to the one in runtime pruning. It has the same architecture as runtime pruning but differs in introducing a fully-connected layer as the encoder before RNN. This agent is also optimized by PPO.

## 3.4 INFERENCE

In inference, the static agent is not required any more because the static pruning strategy does not depend on individual input data points but the full shape of $\mathbf{F}_{in}$. Therefore, the output action $a_t^s$ is fixed to each layer $t$. With the action $a_t^s$ and the rate $R_r$, we can decide which filters can be pruned permanently. Specifically, channels with $((1 - a_t^s)(1 - R_r))$-smallest static importance values are pruned permanently.

## 4 EXPERIMENT

We evaluate our DRL pruning framework on two popular datasets: CIFAR-10 (Krizhevsky, 2009) and ImageNet ILSVRC2012 (Russakovsky et al., 2015), to show the advantage over other channel pruning methods. We analyze the effect of hyper-parameters and different sparsity settings on CIFAR-10. For CIFAR-10, we use M-CifarNet (Zhao et al., 2018) as the backbone CNN. On ImageNet ILSVRC2012, ResNet-18 is used as the backbone CNN.

## 4.1 IMPLEMENTATION DETAILS

We start with a pretrained backbone CNN. Firstly we finetune the backbone CNN and train runtime importance predictor jointly, with sparsity $d_r = 1$ and fixed all static pruning importance $\mathbf{u}_s$ to 1. Then we remove the restriction on the static pruning importance $\mathbf{u}_s$, and train static pruning importance, the backbone CNN and the runtime importance predictor, with sparsity $d_s = 1$ and runtime pruning sparsity fixed as $d_r = 1$. After finetuning, we use the DRL agents to predict the sparsity given computation and storage constraints. The DRL agents and the CNN with runtime/static importance are trained in alternating manner: We first fix the CNN as well as runtime/static importance and train two DRL agents, regarding the CNN as environments. Then we fix two agents and finetune the CNN and runtime/static importance. We repeat these two steps until convergence is achieved. We use Adam optimizer for both DRL agent and CNN, and set learning rate at $10^{-6}$ for the DRL agents. For CNN finetuning and runtime/static importance training, the learning rate is set to $10^{-3}$ on CIFAR-10. On ImageNet ILSVRC2012, the learning rate starts from $10^{-3}$ and is divided by 10 after 15 millions iterations.

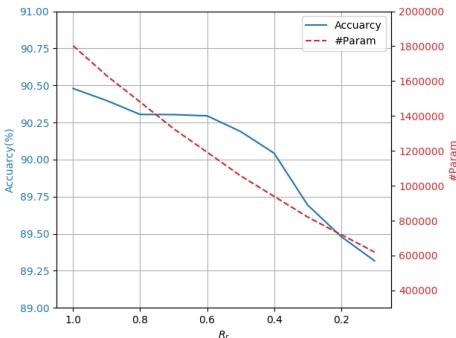 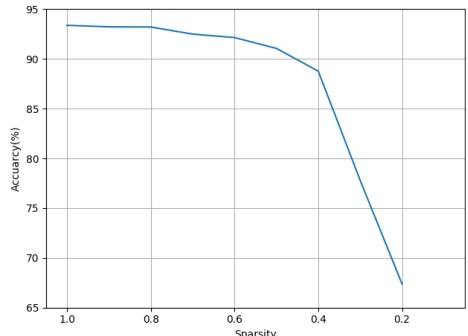

Figure 2: Trade-off between runtime pruning and static pruning at sparsity 0.45. X-axis is the rate $R_r$

Figure 3: Comparison accuracy drop for M-CifarNet on CIFAR-10 with computational budget.

| Method | Baseline top-1 acc. | Top-1 acc. | $\Delta$ top-1 acc. | Baseline top-5 acc. | Top-5 acc. | $\Delta$ top-5 acc. | Speed-up |
|---|---|---|---|---|---|---|---|
| DCP (Zhuang et al., 2018) | 69.64 | 67.35 | -2.29 | 88.98 | **88.86** | **-0.12** | 1.71× |
| FPGM (He et al., 2019) | 70.28 | 68.41 | -1.87 | 89.63 | 88.48 | -1.15 | 1.71× |
| Dynamic Sparse Graph (Liu et al., 2019) | 69.48 | 64.8 | -4.68 | - | - | - | 1.4 × |
| CGNN (Hua et al., 2018) | 69.02 | 67.95 | -1.07 | 88.84 | 88.21 | -0.63 | 1.63× |
| FBS (Gao et al., 2019) | 70.71 | 68.17 | -2.54 | 89.68 | 88.22 | -1.46 | **1.98**× |
| Ours ($R_r = 0.5$) | 69.758 | **68.79** | **-0.968** | 89.078 | 88.534 | -0.544 | 1.94× |

Table 3: Comparison with the state-of-the-art channel pruning ResNet-18 on ImageNet. Speed-up is calculated on MACs.

## 4.2 EXPERIMENTAL RESULTS ON CIFAR-10

We compare our proposed method with the following state-of-the-art runtime pruning methods: FBS (Gao et al., 2019), RNP (Lin et al., 2017) on CIFAR-10. The comparison results at sparsity 0.5 and 0.7 are shown in Table 1 and Table 2 respectively. Note that for fair comparison with other methods, the computation and storage budget constraints in our method is calculated according to the sparsity of other methods. Under these constraints, our method does not necessarily lead to the same sparsity as other methods in each layer. RNP cannot set exact sparsity ratio. Instead, its average sparsity ratio is accessible only during testing, which is 0.537 in Table 1. The result of FBS is reproduced using the released code[1]. The column *#Params* represents the number of parameters compared to the backbone CNN.

Table 1 shows that our method outperforms other state-of-the-art methods, achieving highest accuracy at an overall sparsity ratio of 0.5. Our method has very close computation speed-up compared to FBS, but outperforms FBS around 0.48% to 0.76%. When the runtime pruning strategy is solely considered by setting $R_r = 1$, our method surpasses other comparison methods, indicating that our DRL-based framework improves the performance of channel runtime pruning. By balancing runtime and static pruning via setting $R_r = 0.5$, our method reduces the number of the overall stored parameters and achieves lower accuracy drop than other methods. Table 2 shows that our method outperforms FBS at sparsity of 0.7. When $R_r = 0.5$, our method achieves better performance than the baseline CNN with 2× speed-up and contains less parameters.

We also study the relation between $R_r$ and network compactness in our framework. Fig. 2 demonstrates the impact of $R_r$ when sparsity is 0.45. The hyper-parameter $R_r$ determines how much we trust about runtime pruning. With $R_r$ close to 1, the accuracy becomes higher due to the more dynamic network flexibility but the space of the parameters storage also increases. When $R_r$ diminishes, the network accuracy decreases but the parameter storage is reduced.

Fig. 3 shows the performance of various sparsity ratios in our method. Again, our method does not prune with one single sparsity ratio for all layers, but uses the sparsity ratio to calculate computation and storage constraints, with which the sparsity ratio is learned for each layer. Fig. 3 demonstrates

---

[1]https://github.com/deep-fry/mayo

that our method holds the accuracy when sparsity is larger than about 0.5, which corresponds to about $4\times$ computational acceleration.

### 4.3 EXPERIMENTAL RESULTS ON IMAGENET ILSVRC2012

We compare our method with state-of-the-art channel pruning methods on ImageNet ILSVRC2012 as shown in Table 3. In this experiment, we use ResNet-18 as the backbone CNN. Among the state-of-the-art pruning methods for comparison, FBS (Gao et al., 2019) and CGNN (Hua et al., 2018) are runtime pruning methods. The overall sparsity ratio of our method is 0.7, which is under the same setting of FBS. Our method with $R_r = 0.5$ achieves the smallest top-1 accuracy drop compared with other methods, and also achieves the highest top-1 accuracy after pruning. Overall, our proposed method achieves comparable or better performance compared to other methods with more acceleration. Our method has very close MACs to FBS, while the number of preserved parameters is reduced to $81.2\%$ of the baseline.

## 5 CONCLUSION

In this paper, we present a deep reinforcement learning based framework for deep neural network channel pruning in both runtime and static sheme. Specially, channels are pruned according to input feature as runtime pruning, and based on entire training dataset as static pruning, with 2 reinforcement agents to determine the corresponding sparsity. Our method combines the merits of runtime and static pruning, and provides trade-off between storage and dynamic flexibility. Extensive experiments demonstrate the effectiveness of our proposed method.

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

## A ADDITIONAL EXPERIMENTAL RESULTS

### A.1 COMPARISON TO SEPARATELY STATIC AND RUNTIME PRUNING

In this section, we compare our method with two additional baseline methods. One is a variation of our method by separately training static pruning and the training runtime pruning. In this method, we start from a pretrained backbone CNN, $f(\cdot)$ and $u_s$. Then we add the static DRL agent to prune channels statically by learning the static policy and $u_s$. Finally, we add the runtime DRL agent to prune channels dynamically, by fixing the static DLR agent and $u_s$, and updating the runtime DLR agent and $f(\cdot)$ only. Another method is to combine state-of-the-art static and runtime pruning methods. We start from a pretrain backbone CNN, and then prune channels with the static pruning method FPGM (He et al., 2019), and finally prune channels with the runtime method FBS (Gao et al., 2019). The experimental results are shown in Table4.

| Method | Baseline acc. | Acc. | $\Delta acc.$ |
|---|---|---|---|
| ours (runtime only $R_r = 1$) | 92.07 | 91.333 | **-0.737** |
| ours ($R_r = 0.5$) | 92.07 | 91.066 | **-1.004** |
| ours (separate static and runtime) | 92.07 | 90.965 | -1.0105 |
| FPGM+FBS | 92.07 | 90.456 | -1.614 |

Table 4: Comparison to methods with separately static and runtime pruning on Cifar-10 at sparsity 0.5.

## A.2 STORAGE/ACCURACY TRADE-OFF

Besides Fig. 2, to further illustrate the trade-off between storage and accuracy, we show additional results in Fig. 4.

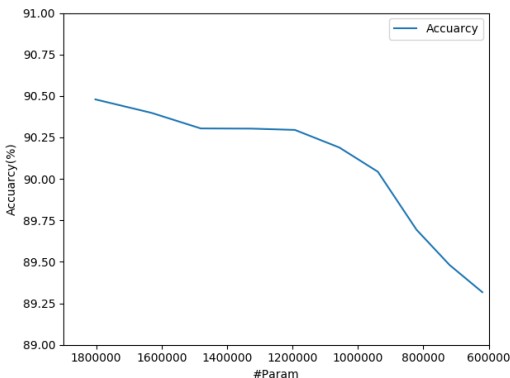

Figure 4: Trade-off between runtime pruning and static pruning at sparsity 0.45. X-axis is storage(number of parameters), Y-axis is accuracy.

## B PSEUDOCODE OF TRAINING PROCESS

---
**Algorithm 1:** Training process

---
**INPUT:** pretrained backbone CNN, computation budget $\bar{B}_{com}$, storage budget $\bar{B}_{param}$
**OUTPUT:** backbone CNN, importance predictor $f(\cdot)$, static pruning importance $u_s$, runtime and static DRL agents

Add runtime importance predictor $f(\cdot)$ and static pruning importance $u_s$.;
$u_s \leftarrow \mathbf{1}$;
$d_s \leftarrow 1$;
$d_r \leftarrow 1$;
**while** *not converge* **do**
   |   fix $u_s$, update $f(\cdot)$ and backbone CNN;
**end**
**while** *not converge* **do**
   |   update $u_s$, $f(\cdot)$ and backbone CNN;
**end**
add runtime DRL agent and static DRL agent to predict actions $a_t^r$ and $a_t^s$, and alternate $d_r$ and $d_s$ at each layer $t$ ;
**while** *not converge* **do**
   |   **for** $i \leftarrow 1$ **to** $N_1$ **do**
   |    |   Forward entired model;
   |    |   Compute rewards $R_t^r$ and $R_t^s$ using budget $\bar{B}_{com}$ and $\bar{B}_{param}$ ;
   |    |   Fix $u_s$, $f(\cdot)$ and backbone CNN, update runtime and static DRL agents by PPO loss;
   |   **end**
   |   **for** $i \leftarrow 1$ **to** $N_2$ **do**
   |    |   Fix runtime and static DRL agents, update $u_s$, $f(\cdot)$ and backbone CNN by cross entropy
   |    |   loss
   |   **end**
**end**

---

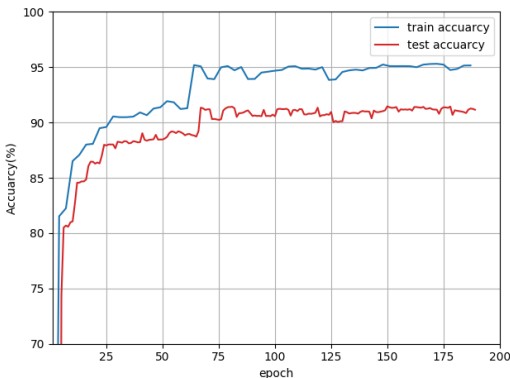

Figure 5: Training curve of our method on CIFAR-10 at sparsity 0.5 with $R_r = 0.5$.

## C  TRAINING CURVE

We show the training curve of our method in Fig. 5. We train our method on CIFAR-10 at sparsity 0.5 with $R_r = 0.5$. The figure includes the accuracy curves evaluated on training set and test set, showing that our methods is stable during training.

