# OpenReview forum: "Storage Efficient and Dynamic Flexible Runtime Channel Pruning via Deep Reinforcement Learning"
_ICLR.cc/2020/Conference — Reject_

### Official Review · AnonReviewer2 · 2019-10-24
**Official Blind Review #2**

**Rating:** 3

**Review:**

This paper presents an offline and online pruning method for CNNs where RL is used for tuning the sparsity.

Pruning methods are important for real-time applications of CNNs on low resourced devices, so the paper addresses a genuine need. However, the paper is poorly written and the experimental results are not as convincing.

1) Majority of the description of the models and architecture is written in text and is very difficult to parse, while this could've been avoided by usage of mathematical notations for operations. This is specifically evident for early parts of sec 3 which makes parsing and reading very difficult.

2) It's not clear why the baseline performances in the experimental section are different across different methods. If I understand correctly, all the original non-pruned models should be the same for this experiment to make sense. This is specially important for Tab 2 where the closest competing algorithms (FBS and CGNN). Due to using different baselines, the numbers are not comparable.

3) One thing that is not clear in the text is the computation taken by the runtime pruner architecture. Are these rolled in the estimations on the speedup?

**Experience Assessment:**

I do not know much about this area.

**Review Assessment: Checking Correctness Of Derivations And Theory:**

I assessed the sensibility of the derivations and theory.

**Review Assessment: Checking Correctness Of Experiments:**

I carefully checked the experiments.

**Review Assessment: Thoroughness In Paper Reading:**

I read the paper at least twice and used my best judgement in assessing the paper.

---

> ### Author Response · Authors · 2019-11-15
> **Response to Review #2**
>
> Thanks for the constructive comment. We would like to address your concerns as follow.
>
> Q1:	“1) Majority of the description of the models and architecture is written in text and is very difficult to parse, while this could've been avoided by usage of mathematical notations for operations. This is specifically evident for early parts of sec 3 which makes parsing and reading very difficult.”
> R1: Thanks for your suggestion. We have revised the typos in our original submission during rebuttal. However, due to the rule of ICLR, we cannot significantly revise our paper during rebuttal period. We will carefully rewrite this part in the final version. We have added pseudocode in appendix B to illustrate the training process. Besides, we would like to summarize the training process presented in Sec. 3 in the following steps:
> (1) At the beginning, we have the pretrained backbone CNN, i.e., M-CifarNet on CIFAR-10 or ResNet-18 on ImageNet.
> (2) We add the runtime importance predictor $f(\cdot)$ and the static pruning importance $u_s$ for each convolutional layer of the backbone CNN. Eq.6 and Eq.2 are applied on the output of convolution layers.
> (3) We train runtime importance predictor $f(\cdot)$ and static pruning importance $u_s$. We initialize $u_s$ as 1s and randomly initialize the weights of $f(\cdot)$. The sparsity ratios $d_s$ and $d_r$ are fixed to 1. This training process consists of two steps as follows.
> 		a) Fix $u_s$, update $f(\cdot)$ and backbone CNN until converge.
> 		b) Update $u_s$, $f(\cdot)$ and backbone CNN until converge.
> (4) We add the runtime DRL agent and the static DRL agent to the model. They predict actions $a_t^r$ and $a_t^s$ to alternate $d_r$ and $d_s$ at each layer $t$.
> (5) Alternatingly update the DRL agents and other components as shown in the following two steps until convergence is reached
> 		a) Fix $u_s$, $f(\cdot)$ and the backbone CNN, update the runtime and the static DRL agents by PPO loss
> 		b) Fix the runtime and the static DRL agents, update $u_s$, $f(\cdot)$ and backbone CNN by cross entropy loss
>
> Q2:	“2) It's not clear why the baseline performances in the experimental section are different across different methods. If I understand correctly, all the original non-pruned models should be the same for this experiment to make sense. This is specially important for Tab 2 where the closest competing algorithms (FBS and CGNN). Due to using different baselines, the numbers are not comparable.”
> R2: For the reason of using different baseline acc is that different deep learning frameworks are used in this experiment. For example, the released code of FBS (https://github.com/deep-fry/mayo) uses a framework called “mayo” implemented based on TensorFlow, while our approach is implemented on PyTorch. Different frameworks lead to different performance, even with same network architecture and same data. For table 3 of comparison experiment on ImageNet, comparison methods also use different frameworks to implement their approach. For fair comparison, we use PyTorch's officially pretrainined ResNet-18 as our baseline.
> Although the "Baseline acc." is different, it still makes sense to the comparison result. In Table 1, our baseline acc. is higher than FBS. Usually higher pretrained accuracy is easier to drop after pruning. However, our accuracy after pruning is still higher than FBS, and the gap $\delta acc$ is also smaller than FBS.
> Simillary, in the comparison result of ImageNet, the baseline acc of our method is higher than CGNN. The accuracy drop of our method is still smaller than CGNN, and the post-pruning acc is higher than CGNN. Moreover, our post-pruned accuracy and accuracy drop are also better than FBS.
>
> Q3:	“3) One thing that is not clear in the text is the computation taken by the runtime pruner architecture. Are these rolled in the estimations on the speedup?”
> R3: Yes, it has been rolled in the estimations on the speedup.

---

### Official Review · AnonReviewer1 · 2019-10-25
**Official Blind Review #1**

**Rating:** 3

**Review:**

This paper proposes to learn static and dynamic channel pruning policies for convolutional neural networks. Static pruning depends only on the training dataset and is computed once before the model is deployed. Dynamic pruning is input-dependent. The policies are obtained with deep reinforcement learning on the training dataset using a combination of the loss function and storage/computation resource budgets as a reward signal. The key novelty in this paper is to combine static and dynamic pruning which can obtain the benefits from both worlds. Experimentally, the learned pruning policies are competitive with recent dynamic pruning approaches on CIFAR-10 and ILSVRC2012, in terms of both final test accuracy and number of parameters/inference time.

Overall, I do like the idea of combining static and dynamic pruning, the DRL approach is reasonable and it seems to do well in practice. However, there are some key issues that must be addressed by the authors. In summary, these are:

1- Additional baseline methods: why not compare against a simple combination of the best published static pruning method, and one of the best published dynamic pruning methods (e.g. FBS or RNP)? I'd like to understand what the added value of jointly learning the static and dynamic pruning policies is. If a simple combination of existing static and dynamic methods works well, you will need to justify the need for the more complicated DRL approach you propose in this paper.

2- Writing: it has to be substantially improved; please see the sections Writing and Minor below.

3- Training cost: how stable is the training process described in Section 4.1? Does it much longer to train compared to pure dynamic pruning methods?

Related work:
- Discuss connections to routing networks which adaptively route layer outputs to the next layer's modules: Rosenbaum, Clemens, Tim Klinger, and Matthew Riemer. "Routing networks: Adaptive selection of non-linear functions for multi-task learning." arXiv preprint arXiv:1711.01239 (2017).

Writing:
- Sections 3.3 up to Section 5 need lots of rewriting; I suggest some changes in "Minor" below but please make a full pass as the paper is difficult to read at the moment.
- Section 4.1 is extremely difficult to read and so I don't really understand how you train the pruning policies. Please improve the writing and summarize the process in pseudocode or illustrate it. Also, I believe 4.1 should be a section of 3 rather than in experiments. It is extremely important to understand how the network+DRL agents are trained!

Clarification:
- Number of pruned channels, runtime vs static: \ceil{d_r C} vs (C - \ceil{d_s C}); why are these different in form? Seems like the static formula prunes \ceil{(1-d_s) C}. Why is that?
- Tables 1-2: what is the "Baseline acc." and why is it different for each method? Isn't this the accuracy of the same network before any pruning?

Minor:
- Title is too long: particularly the expression "DYNAMIC FLEXIBLE RUNTIME CHANNEL". Perhaps you can think of shorter titles.
- "We consider a standard form of reinforcement learning an agent" --> "We consider a standard form of reinforcement learning: an agent"
- "are the the number of channels" --> "are the number of channels"
- "output feature F_out u_r is predicted" --> "output feature F_out, u_r, is predicted"
- "many existed dynamic" --> "many existing dynamic"
- "Since C_in is various among layers" --> "Since C_in varies among layers"
- "To avoid over-prune the filters and crashed in training, we set a minimum sparsity ratio +α, then the action space change to a_t^r ∈ (+α, 1]." --> "To avoid over-pruning the filters and crashing the training, we set a minimum sparsity ratio +α, such that the action space becomes a_t^r ∈ (+α, 1]."
- "The reward function is proposed to consider both of network accuracy and computation budget." --> "The reward function is considers both the network accuracy and computation budget."
- "and it may be various in scale" --> "and it may vary in scale"
- "large at begin of training and small near converge" --> "large at the beginning of training and small near convergence"
- "which filters can be to prune permanently" --> "which filters can be pruned permanently"
- "on two popular dataset" --> "on two popular datasets"
- "4.1 IMPLEMENT DETAILS" --> 4.1 IMPLEMENTATION DETAILS
- "For CNN fintuning" --> "For CNN finetuning"
- "Noted that for fair" --> "Note that for fair"
- Tables 1-2: "Compare to state-of-the-art" --> "Comparison to state-of-the-art"
- "Our methods has" --> "Our method has"

**Experience Assessment:**

I have read many papers in this area.

**Review Assessment: Checking Correctness Of Derivations And Theory:**

N/A

**Review Assessment: Checking Correctness Of Experiments:**

I carefully checked the experiments.

**Review Assessment: Thoroughness In Paper Reading:**

I read the paper at least twice and used my best judgement in assessing the paper.

---

> ### Author Response · Authors · 2019-11-15
> **Response to Review #1 [1/2]**
>
> Thanks for the constructive comment. We would like to address your concerns as follows.
>
> Q1:	“1- Additional baseline methods: why not compare against a simple combination of the best published static pruning method, and one of the best published dynamic pruning methods (e.g. FBS or RNP)? I'd like to understand what the added value of jointly learning the static and dynamic pruning policies is. If a simple combination of existing static and dynamic methods works well, you will need to justify the need for the more complicated DRL approach you propose in this paper.”
> R1: Thanks for your suggestion. We have added a comparison experiment in Appendix A.1. In this experiment, we compare our method with an additional baseline, which combines the static pruning method FPGM (He et al. 2019) and the dynamic pruning method FBS (Gao et al. 2019).
> For fair comparison, we first use FPGM to prune M-CifarNet with sparsity ratios of 0.75 and then prune M-CifarNet by FBS with sparsity ratios of 2/3. In this way, the final sparsity is (1*0.75*2/3=)0.5, which is the same as the sparsity setting of our method shown in Table 1.
>
> Q2:	“3- Training cost: how stable is the training process described in Section 4.1?”
> R2: We have added a training curve to show the stability of our methods in Fig.5 in Appendix C. This curve shows that our method is stable and easy to train.
>
> Q3:	“Does it much longer to train compared to pure dynamic pruning methods?”
> R3: In our hardware platform, the time of training an epoch on CIFAR10 for updating the DRL agents is about 53s, an epoch for updating backbone CNN, runtime importance predictor $f(\cdot)$ and static pruning importance $u_s$ is about 30s. On the same hardware platform, an epoch of FBS is about 24s. Due to the training procedure of FBS, it first sets sparsity $d=1$ and finetunes the network. Then it reduces sparsity by 10% and finetunes network again. For each new target sparsity, FBS takes 300 epochs to train network. If target sparsity ratios is 0.5, it requires 1800 epochs (approximately 12 hours) to finish training. Our method needs 400 epochs(approximately 3.33 hours) for pretraining runtime importance predictor $f(\cdot)$ and static pruning importance $u_s$. Then our methods directly adapt to target sparsity. It requires 128 DRL agents epochs and 64 backbone CNN epochs, approximately 2.48 hours. Our method needs approximately 5.81 hours of training for a specified sparsity ratio.
>
> Q4:	“Related work:
> - Discuss connections to routing networks which adaptively route layer outputs to the next layer's modules: Rosenbaum, Clemens, Tim Klinger, and Matthew Riemer. "Routing networks: Adaptive selection of non-linear functions for multi-task learning." arXiv preprint arXiv:1711.01239 (2017).”
> R4: C. Rosenbaum et al. proposed a multi-agent method to perform routing inside a neural network. The goal of their work is to learn routing to adjust multi-task learning. Our work performs runtime pruning to accelerate a pretrained CNN, We have included this related work in the latest version of our paper.
>
> Q5:	“- Sections 3.3 up to Section 5 need lots of rewriting; I suggest some changes in "Minor" below but please make a full pass as the paper is difficult to read at the moment.”
> R5: Sorry for the typos and very thankful to your suggestion. We have revised them in the latest version. We also have proofread the paper to revise other typos.

---

> ### Author Response · Authors · 2019-11-15
> **Response to Review #1 [2/2]**
>
> Q6:	“- Section 4.1 is extremely difficult to read and so I don't really understand how you train the pruning policies. Please improve the writing and summarize the process in pseudocode or illustrate it. Also, I believe 4.1 should be a section of 3 rather than in experiments. It is extremely important to understand how the network+DRL agents are trained!”
> R6: We have added pseudocode in appendix B to illustrate the training process. Besides, we would like to explain the training process in the following steps:
> (1) At the beginning, we have the pretrained backbone CNN, i.e., M-CifarNet on CIFAR-10 or ResNet-18 on ImageNet.
> (2) We add the runtime importance predictor $f(\cdot)$ and the static pruning importance $u_s$ for each convolutional layer of the backbone CNN. Eq.6 and Eq.2 are applied on the output of convolution layers.
> (3) We train runtime importance predictor $f(\cdot)$ and static pruning importance $u_s$. We initialize $u_s$ as 1s and randomly initialize the weights of $f(\cdot)$. The sparsity ratios $d_s$ and $d_r$ are fixed to 1. This training process consists of two steps as follows.
> 		a) Fix $u_s$, update $f(\cdot)$ and backbone CNN until converge.
> 		b) Update $u_s$, $f(\cdot)$ and backbone CNN until converge.
> (4) We add the runtime DRL agent and the static DRL agent to the model. They predict actions $a_t^r$ and $a_t^s$ to alternate $d_r$ and $d_s$ at each layer $t$.
> (5) Alternatingly update the DRL agents and other components as shown in the following two steps until convergence is reached
> 		a) Fix $u_s$, $f(\cdot)$ and the backbone CNN, update the runtime and the static DRL agents by PPO loss
> 		b) Fix the runtime and the static DRL agents, update $u_s$, $f(\cdot)$ and backbone CNN by cross entropy loss
>
>
> Q7:	“- Number of pruned channels, runtime vs static: \ceil{d_r C} vs (C - \ceil{d_s C}); why are these different in form? Seems like the static formula prunes \ceil{(1-d_s) C}. Why is that?”
> R7: Sorry for this mistake. It should be $(C - \lceil{d C}\rceil)$ for both runtime and static. We define $d$, including $d_s$ and $d_r$, as the ratio of remaining filters over all filters. The numbers of pruned channels should be $(C - \lceil{d_s C}\rceil)$ and $(C - \lceil{d_r C}\rceil)$ respectively. These two terms have been revised in the latest version.
>
> Q:8	“- Tables 1-2: what is the "Baseline acc." and why is it different for each method? Isn't this the accuracy of the same network before any pruning?”
> R8: "Baseline acc." is the accuracy before any pruning. The reason of different "Baseline acc." is that different deep learning frameworks are used in this experiment. For example, the released code of FBS (https://github.com/deep-fry/mayo) uses a framework called “mayo” implemented based on TensorFlow, while our approach is implemented on PyTorch. Different frameworks lead to different performance, even with same network architecture and same data. For table 3 of comparison experiment on ImageNet, comparison methods also use different frameworks to implement their approach. For fair comparison, we use PyTorch's officially pretrainined ResNet-18 as our baseline.
> Although the "Baseline acc." is different, it still makes sense to the comparison result. In Table 1, our baseline acc. is higher than FBS. Usually higher pretrained accuracy is easier to drop after pruning. However, our accuracy after pruning is still higher than FBS, and the gap $\delta acc$ is also smaller than FBS.

---

### Official Review · AnonReviewer3 · 2019-11-04
**Official Blind Review #3**

**Rating:** 6

**Review:**

This work introduces a Reinforcement Learning based framework that simultaneously learns both a static and dynamic pruning strategy. The combination allows the static pruner to decrease the required storage while the dynamic pruning can optimize the required compute using input-dependent pruned weights. The RL agent can dynamically learn the optimal sparsity distribution among the different layer of the networks while staying under a resource constraint as opposed to other methods which often enforce a layer level sparsity ratio. It demonstrates the efficacy of the algorithm on CIFAR10 and ILSVRC2012 and showed the effect of the tradeoff between static and dynamic pruning.

Overall I believe this paper is a borderline accept. It proposes a unified framework to manage the trade-off between static pruning to decrease storage requirements and network flexibility for dynamic pruning to decrease runtime costs. The empirical results demonstrate the capability of the framework but would benefit from some clarification and additional ablations.

Pros:
Proposed an RL formulation of a unified framework for static and dynamic channel pruning.

The empirical results demonstrate the ability of the model to achieve high accuracy while sparsifying the compute and balancing storage consumption and accuracy. Strong results are shown for CIFAR10 and ILSVRC2012.

Demonstrated a tradeoff between static pruning and runtime-pruning through ablation of R_r.

Cons:
It is unclear how speed-up calculated. Is it wall clock and benchmarked on what device? What was the cost of running the RL agent during runtime?
If you are computing MACs, that should be reported as such unless a strong correlation can be proven with the particular pruning scheme. MAC reduction does not translate directly to speedup in hardware.

It could be clearer if the ablation of R_r also demonstrated a storage/accuracy tradeoff.

It is unclear if similar results may be achieved by first running a static pruning and then separately training a dynamic pruning algorithm on the already statically pruned network?
It might benefit from an ablation study with and without simultaneously training static pruning?

Some of the algorithmic details could benefit from some clarification.
In section 3.2, it is unclear to me the effect of R_r during training. It seems that the agent could learn to over select channels to prune to adapt to R_r. Section 3.4 seems to lack inclusion of R_r in the number of statically pruned values. The treatment of M_0 in section 3.2 separate from  M_r and M_s seems to make the number of statically pruned filters during training depend on the layers chosen by dynamic pruning. It seems like it may cause differing sparsity between training and inference time.

**Experience Assessment:**

I have published one or two papers in this area.

**Review Assessment: Checking Correctness Of Derivations And Theory:**

I assessed the sensibility of the derivations and theory.

**Review Assessment: Checking Correctness Of Experiments:**

I assessed the sensibility of the experiments.

**Review Assessment: Thoroughness In Paper Reading:**

I read the paper at least twice and used my best judgement in assessing the paper.

---

> ### Author Response · Authors · 2019-11-15
> **Response to Review #3**
>
> Thanks for the constructive comment. We would like to address your concerns as follow.
>
> Q1:	“It is unclear how speed-up calculated. Is it wall clock and benchmarked on what device? What was the cost of running the RL agent during runtime? If you are computing MACs, that should be reported as such unless a strong correlation can be proven with the particular pruning scheme. MAC reduction does not translate directly to speedup in hardware.”
> R1: The “speed-up” mentioned in Table 1, 2 and 3 are calculated on MACs. We have clarified it in the latest version of our paper. The computation cost of running RL agent in inference is tiny compared to the cost of the convolutional layer. We use global pooling to extract features from the feature map, and then feed them into a fully-connected encoder to produce a state $s_t^r$, which is only of 128 dimensions. The state $s_t^r$ is fed into a simple RNN of 128 dimensions, followed by a fully-connected layer to produce a scalar action. The MACs of the RL agent is 0.21% of the pretrained network.
>
> Q2:	“It could be clearer if the ablation of R_r also demonstrated a storage/accuracy tradeoff.”
> R2: We have added Fig. 4 in Appendix to further demonstrate the storage/accuracy tradeoff.
>
> Q3:	“It is unclear if similar results may be achieved by first running a static pruning and then separately training a dynamic pruning algorithm on the already statically pruned network? It might benefit from an ablation study with and without simultaneously training static pruning?”
> R3: To address this issue, we have added a new comparison experiment with the method that first statically prunes channels, and then dynamically prunes the rest channels. The experimental results are presented in Appendix A.1. It shows that our simultaneously training strategy achieves better performance than the separately training strategy.
>
> Q4.	 “In section 3.2, it is unclear to me the effect of R_r during training. It seems that the agent could learn to over select channels to prune to adapt to R_r. Section 3.4 seems to lack inclusion of R_r in the number of statically pruned values. The treatment of M_0 in section 3.2 separate from  M_r and M_s seems to make the number of statically pruned filters during training depend on the layers chosen by dynamic pruning. It seems like it may cause differing sparsity between training and inference time.”
> R4: Sorry for unclear explanation in the original submission. Before we explain the effect of $R_r$ here, we would like to correct a term in section 3.4:
> “Channels with $((1-a^s_t)/2)-smallest$ …”  -->  “Channels with $((1-a^s_t)(1-R_r))-smallest$ … ”
> $R_r$ does NOT cause different sparsity between training and inference time. $R_r$ is the rate denoting how much we trust the pruning result from runtime pruning and $1-R_r$ is about how much we trust from static pruning. $R_r \in [0,1]$ is a hyperparameter which will be fixed during training and inference. $M_O$ is not the final pruning result, instead, it is the overlap/interset of $M_r$ and $M_s$. The final pruning result is $M$ in Eq. 5. Since the DRL agents have same behaviors in training and inference, and $R_r$ is fixed, sparsity remains the same between training and inference time.

---

### Decision · Program_Chairs · 2019-12-19

**Decision:**

Reject

**Comment:**

Main content: Proposes a deep RL unified framework to manage the trade-off between static pruning to decrease storage requirements and network flexibility for dynamic pruning to decrease runtime costs
Summary of discussion:
reviewer1: Reviewer likes the proposed DRL approach, but writing and algorithmic details are lacking
reviewer2: Pruning methods are certainly imortant, but there are details missing wrt the algorithm in the paper.
reviewer3: Presents a novel RL algorithm, showing good results on CIFAR10 and ISLVRC2012. Algorithmic details and parameters are not clearly explained.
Recommendation: All reviewers liked the work but the writing/algorithmic details are lacking. I recommend Reject.